# Toward Soil Nutrient Security for Improved Agronomic Performance and Increased Resilience of Taro Production Systems in Samoa

Diogenes L. Antille [1],*, Ben C. T. Macdonald [1], Aleni Uelese [2], Michael J. Webb [3], Jennifer Kelly [1], Seuseu Tauati [2], Uta Stockmann [1], Jeda Palmer [4] and James R. F. Barringer [5]

1. CSIRO Agriculture and Food, Black Mountain Science and Innovation Precinct, Canberra, ACT 2601, Australia
2. Scientific Research Organisation of Samoa, Apia P.O. Box 6597, Samoa
3. CSIRO Agriculture and Food, Townsville, QLD 4811, Australia
4. CSIRO Agriculture and Food, Brisbane, QLD 4067, Australia
5. Landcare Research-Manaaki Whenua, Lincoln 7608, New Zealand
* Correspondence: dio.antille@csiro.au; Tel.: +61-(02)-6218-3835

**Abstract:** A progressive decline in soil fertility in taro (*Colocasia esculenta* L., Schott) production systems has contributed to reduced crop productivity and farm profitability, and is recognized to be a threat to soil nutrient and food security in Samoa. Evidence based on three years of field experimentation showed that appropriate nutrient budgeting is required to reduce soil nutrient deficits and mitigate soil organic carbon loss. Balanced crop nutrition coupled with appropriate crop husbandry can significantly improve productivity and narrow yield gaps. A framework to guide nutrient recommendations for taro production systems is presented and discussed. This framework proposes that recommendations for N be derived from the yield-to-N response function (from which the most economic rate of N can be estimated) and that for other nutrients, namely P, K, Ca, and Mg, recommendations be based on replacement. The replacement strategy requires the development of soil nutrient indexes, which can be used to define the long-term nutrient management policy at the field scale. This long-term policy is informed by soil analyses, and it will determine whether existing soil nutrient levels are to be maintained or increased depending on the focus (productivity, profitability, environmental protection). If soil nutrients were already at an agronomically satisfactory level, their application may be omitted in some years to help reduce crop production costs, improve use efficiency, and ensure environmentally safe levels in soil are not exceeded.

**Keywords:** fertilizer recommendations; legume intercropping; nutrient budgeting; soil nutrient balance; soil nutrient index; yield gap

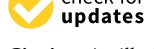



## 1. Introduction

Pacific Island nutrient cycles have been increasingly modified since human settlement, which in Remote Oceania (including Fiji, Samoa, and Tonga) commenced more than 3000 years ago [1]. Agricultural intensification in Pacific countries has resulted in further changes in the islands' nutrient flows [2]. In Samoa, declined soil fertility associated with nutrients exported from cropped land represents a major production constraint, and in some cases, it is considered to be the primary cause of the yield gap [3] (yield gap is the difference between the attainable yield and on-farm, actual, yield [4]). The loss of soil fertility on cultivated land is exacerbated by a concurrent decline in soil organic carbon (SOC) [5]. The imbalance in nitrogen (N), phosphorus (P), and potassium (K) is only one aspect of the plant nutrition and should be used to highlight the need to adequately understand nutrient flows, soil fertility management, and nutrient budgeting in tropical agricultural systems and in doing so ensuring that SOC levels are, at least, maintained [6,7].

In Samoa, taro (*Colocasia esculenta* L., Schott) is a major dietary staple and an important source of export earnings [8]. Taro crops are planted between late August and November, immediately before the rainy season, and they are harvested between 6 and 11 months after planting [9]. Taro growers, especially those who produce it for subsistence, rarely use any fertilizers or soil amendments, and some growers claim the application of nutrients adversely affects the quality and taste of the corm. Despite this, taro has been reported to respond well to the application of mineral fertilizers and soil amendments, such as compost and manures [10]. Research has shown that the use of soil amendments in taro production is economically viable and improves the nutritional value (protein and micronutrient content) of the corm [11,12]. Much of the past research has given little or no consideration to quantifying nutrient balances and linking soil nutrient deficits with yield gaps in Samoan taro production systems, with some exceptions (e.g., [13]). However, agronomic information derived from experimental observations in earlier studies was never translated into practical nutrient management guidelines or fertilizer recommendations for those systems. Achieving sustained productivity increases to narrow existing yield gaps requires identification, and subsequent adoption by growers, of technically and economically viable nutrient management practices. Therefore, this work seeks to demonstrate that current agronomic practices for taro production could be improved with a few adjustments to the way crop nutrition is managed. To meet growers' aspirations of narrowing yield gaps by lifting productivity and societal expectations of improving soil and food security, these adjustments must be implemented without increasing the environmental footprint of taro cropping.

The objectives of the work reported in this article were to (*i*) determine taro yield for crops managed under a range of widely adopted agronomic practices and be able to quantify yield gaps, (*ii*) provide estimates of soil nutrient balances, and (*iii*) develop a conceptual framework to guide nutrient recommendations for taro production systems. To achieve these objectives, field scale experiments were conducted over three years at two locations on Upolu Island, Samoa. The effects of crop nutrition on the soil nutrient balance, taro yield, and nutrient recovery in harvested plant material (corm) were investigated. Soil and agronomic data derived from these experiments were used to determine yield gaps, inform the nutrient management framework presented in this work, and develop a set of recommendations for future research. Experimental treatments included (*i*) standard practice (no nutrients applied), (*ii*) legume intercropping using *Mucuna pruriens* (L., DC.) and *Erythrina subumbrans* (Hassk., Merr.), (*iii*) the application of compound fertilizer that contained N, P, K, and sulfur (S), and (*iv*) the application of a composted material based on poultry manure. An improved understanding of soil nutrient dynamics and the development of nutrient management guidelines for taro production systems will improve farming systems' resilience and soil nutrient security and, therefore, food security in Samoa.

## 2. Materials and Methods

### 2.1. Climate

The climate for Upolu is tropical, hot, humid, and rainy throughout the year, with relative maximum rainfall from December to March, and minimum from June to September. The mean annual rainfall is 2800 mm. The mean number of rainy days per year is 169, and it ranges from 8 days in July to 20 days in January. Rainfall occurs in the form of downpours or thunderstorms, which are often intense but usually short-lived, except between December and March, when rainfall duration increases. Temperatures vary little throughout the year, and they are slightly warmer between December and April compared to the period May–November. On average, the thermal amplitude between day and night is about 6 °C, with night temperatures typically above 20 °C (mean minimum and mean maximum temperatures are 23.7 and 29.7 °C, respectively). Mean relative humidity is approximately 80%.

### 2.2. Experimental Sites

Field trials were established at the Crop Research and Development Station of the Samoan Ministry of Agriculture and Fisheries at Nu'u (13°49.829′ S, 171°50.193′ W, an elevation of 71 m above sea level) in August 2018 (henceforth Nu'u 1) and December 2020 (henceforth Nu'u 2), respectively. A remote site was also established at a commercial farm in southern Upolu (14°00.432′ S, 171°39.492′ W, an elevation of 181 m above sea level) in September 2018 (henceforth Faleālili). Before the experiments were established, both Nu'u 1 and 2, and Faleālili sites had been used for taro production. The two locations are shown in Figure 1.

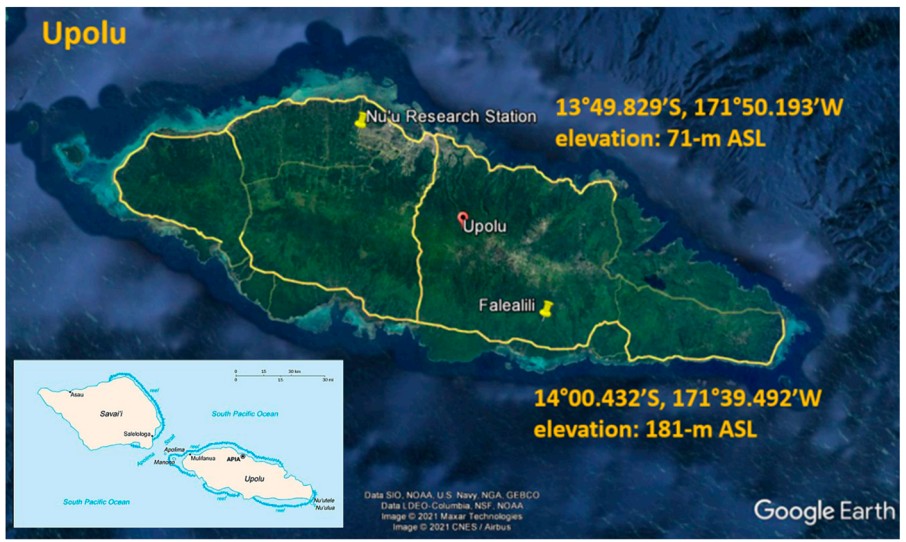

**Figure 1.** A map of Upolu Island (Samoa) showing the locations of the experimental sites established in 2018 and 2020.

The trials at Nu'u 1 and Faleālili were laid out in a completely randomized block design (*n* = 4) and were established to compare the agronomic performance of taro without (control) and with intercropped legumes (*Mucuna* sp. and *Erythrina* sp.). The taro variety used at all sites was Samoa II, which is resistant to taro leaf blight (*Phytophthora colocasiae* Racib.) [14]. The trial sites had 12 plots (plot's dimensions: 7 m long by 6 m wide) and there was a 0.5 m buffer between plots. Within each plot, there were 42 taro plants, established using a standard 1 m × 1 m planting system [15]. At Nu'u 1, taro was planted on 27 August 2018 and harvested on 12 March 2019, and at Faleālili, it was planted on 21 September 2018 and harvested on 7 May 2019. Both legumes were established simultaneously with the taro crops to test their ability to supply N to the crop and consequently reduce the reliance on exogenous N applications (e.g., synthetic fertilizers, and organic amendments). Throughout the season, the legume plants were trimmed as needed to avoid any interference with adjacent plots or treatments. Any overgrown part of the legume plants that extended beyond the boundaries of the plots was cut away and mulched down on the same plot to ensure no additional nutrients were removed from the designated plot area. At Nu'u 2, the trial was also laid out in a completely randomized block design (*n* = 3) to compare the agronomic performance of taro with and without the addition of soil amendments, namely a compound fertilizer commercially known as Blaukorn Classic® (12% N, 8% $P_2O_5$, 15% $K_2O$, 3% $SO_3$), a blend of composted poultry manure ($\approx \frac{3}{4}$ by weight) and malt waste, desiccated coconuts, and coral chips ($\approx \frac{1}{4}$ by weight) with an overall composition of 11% N, 5% $P_2O_5$, 8% $K_2O$, 5% $SO_3$, and 3.5% MgO. In situations where taro crops are supplied with external nutrients, these types of amendments are common, albeit infrequent. At Nu'u 2, a split plot design was then used to compare taro yield and nutrient use efficiency as affected by amendment type and placement (surface application vs. incorporated). For the surface application treatment, amendments were spread around, and in the proximity (<150 mm)

of, the taro plants. For the incorporation treatment, amendments were placed beneath the taro plants in the hole dug by the field operator at planting (at approximately 150 mm below the soil surface). A thin layer of soil was then added on top of the amendment to avoid direct contact between this and the taro plant. Both amendments were applied at planting at a standard rate of 50 g of product (fresh weight basis) per plant, which equated to the 'half-hand full' rate used by local farmers. Therefore, the application rate used at Nu'u 2 represented the standard (local) agronomic practice, and the planting system was the same as that described for Nu'u 1 and Faleālili.

*2.3. Soil Measurements and Analyses*

Soil chemical analyses were conducted at the SROS Laboratories (The Scientific Research Organization of Samoa at Apia, https://sros.org.ws/ (accessed on 26 October 2022)) prior to the experiments using the methods adopted by SPACNET (The South Pacific Agricultural Chemistry Laboratory Network) [16]. Sub-samples from Nu'u 1 and Faleālili were sent to the CSIRO (Commonwealth Scientific and Industrial Research Organisation) Soil Physics Laboratory at Canberra (Australia) for determination of the soil water characteristics. This information was subsequently used to quantify the plant available water capacity (PAWC) of the soils and determine the feasibility of growing taro outside the normal window for the crop. It was postulated that sufficient soil water storage capacity, coupled with soil management practices conducive to increasing rainfall conservation during the dry season (e.g., stubble retention, minimal soil disturbance), could allow a second taro crop to be simultaneously grown (relay intercropped) with the main taro crop. Such an agronomic strategy could potentially increase productivity in a shorter timeframe compared with growing two taro crops sequentially.

For soil water characterization, the relationship between soil water content and water potential was determined on intact soil cores with dimensions of 50 mm diameter by 50 mm long. The cores were manually taken for the middle point of three depth intervals (0–150, 150–300, and 300–600 mm). Gravimetric soil water contents were determined at 0.1, 10, 30, 50, 100, 340, and 600 cm tensions using ceramic suction plates, and subsequently at 15 bar using a pressure plate apparatus, as described by Cresswell [17]. The laboratory determination of the drained upper limit (DUL) and crop lower limit (LL) was subsequently approximated by soil water contents measured at potentials of 100 cm ($DUL_{100}$) and 15 bar ($LL_{15}$), respectively. The difference between $DUL_{100}$ and $LL_{15}$ provides the laboratory determination of PAWC for each depth interval [17]. When equilibration was reached, defined as a change in soil mass <0.05 g over a 24 h period, the soil cores were removed from the plates, weighed, and returned to the plates where the process was repeated for successive water potentials. After the 15 bar measurement was completed, the soil cores were placed in an oven at 105 °C for 72 h to determine the gravimetric water content at each incremental tension. The gravimetric water content was then expressed volumetrically by multiplying it by the soil bulk density ($\rho_b$). The van Genuchten [18] functions were fitted to measured data to describe the relationship between soil water content (expressed volumetrically) and water potential (Equations (1)–(3)). The van Genuchten model parameter $\alpha$ and exponent $\eta$ were estimated, as described by Ngo-Cong et al. [19]. Soil bulk density was determined, as per Blake and Hartge [20], by taking cores of 40 $cm^3$ volume. The soil in the cores was then weighed, placed in an oven at 105 °C for 72 h, and re-weighted for determination of dry weight and gravimetric soil water content. Dynamic changes in volume resulting from changes in water content were negligible and so the volume of soil used for density calculations was equivalent to the volume of the cylinder.

$$S_e = \left[ \frac{1}{1 + (\propto h)^n} \right]^m \tag{1}$$

where

$$m = 1 - \frac{1}{n} \tag{2}$$

and

$$S_e = \frac{(\theta - \theta_r)}{(\theta_S - \theta_r)} \therefore \theta = \theta_r + (\theta_S - \theta_r) \times S_e \qquad (3)$$

where $Se$ is effective saturation, $h$ is the pressure head (cm), $\theta$ is the soil water content, $\theta_S$ and $\theta_r$ are the saturated and residual water contents (all in cm$^3$ cm$^{-3}$), respectively, and $\alpha$ (cm$^{-1}$) and $\eta$ (dimensionless) are fitting parameters that describe the shape of the water retention function.

Unconfined, saturated infiltration rates were measured in the field with the double-ring infiltrometer method [21]. Infiltration rates were obtained by differentiating Kostiakov's function (Equation (4)) with respect to time to describe the relationship between the rate of infiltration and time (Equation (5)) [22]. At all sites, the soils were classified as well-drained [23].

$$F_t = a \times t^n \qquad (4)$$

$$I_t = a \times n \times t^{n-1} \qquad (5)$$

where $F_t$ is cumulative infiltration (mm) at time $t$ (h), $a$ and $n$ are constants, and $I_t$ is the instantaneous water infiltration rate (mm h$^{-1}$).

*2.4. Crop Measurements and Analyses*

Taro yield was determined by removing and weighing the eight central plants from each plot. The corms were then separated from the rest of the plant and weighed, and the yield was expressed as kg of dry matter (DM) per ha (the average water content of the taro corms was $35.67 \pm 4.509\%$, *w/w*). Corm and total plant biomass were used to determine the harvest index expressed as a percentage [24,25]. Nutrient off-take was estimated using elemental (N, P, K, calcium: Ca and magnesium: Mg) concentrations in corm measured in previous experiments at the sites that used the same varieties and reported similar yields [13,26]. The corms were removed from the field at harvest while the rest of the plant biomass was returned to the designated plots. Therefore, nutrient off-take equates to the corm biomass (DM basis) multiplied by its elemental nutrient concentration and is expressed as kg (element) per ha [27–29]. The field scale nutrient balance was estimated from the difference between nutrient inputs (e.g., applied nutrients in amendments, such as compost and fertilizer, and via N fixation in legume intercropped taro) and nutrient outputs (off-take in corm). Nutrient inputs for the controls were assumed to be zero and so the nutrient balance was always negative (net off-take). For legume intercropped taro, it was considered that in the year of establishment the legume plants would contribute about 50 kg N ha$^{-1}$ per year. Reported N supply rates from intercropped legumes in taro crops ranged between 40 and 180 kg N ha$^{-1}$ per year [30–32]. The rationale for choosing a N supply rate within the lower range of previously reported values was consistent with the stage of legume plants development at the sites, and the fact that only about one-third to half the area between taro rows was covered by legumes (as determined by visual assessment at harvest).

Differences in yield between amendment-treated or legume intercropped taro and controls, relative to nutrients supplied as an amendment or via N fixation, were used to denote the agronomic efficiency (AE), as shown in Equation (6) (after [24,29]):

$$AE = \frac{Y_{F \neq 0} - Y_{F=0}}{Rate} \qquad (6)$$

where *AE* is agronomic efficiency (kg kg$^{-1}$), $Y_{F \neq 0}$ and $Y_{F=0}$ are DM yields of amended-treated or legume intercropped taro and the control, and the rate is the amount of given nutrient supplied as an amendment or via N fixation, respectively (all in kg ha$^{-1}$).

*2.5. Statistical Analyses*

Statistical analyses for all measured soil and agronomic variables used GenStat Release 16th Edition [33] and involved analysis of variance (ANOVA). The least significant

differences (LSD) were used to compare means with a probability level of 5%. Statistical analyses were graphically assessed by means of residual plots and normalization of data was not required.

## 3. Results

### 3.1. Soils Measurements and Analyses

Results derived from the baseline characterization of the soils from the experimental sites are presented in Table 1. Overall, there were no statistical differences before vs. after harvest or between the control vs. treatments (legumes) in any of the soil parameters analyzed. The van Genuchten [18] model parameters, used for representing the relationship between soil water content (expressed volumetrically) and water potential are presented in Table 2.

**Table 1.** Baseline characterization of soils at the experimental sites. Notation: $\rho_b$, (dry) soil bulk density; EC, the electrical conductivity of soil; SOC, soil organic carbon; N, nitrogen; P, phosphorus; K, potassium. Depth range: 0–150 mm. For particle size analysis, the soil was sieved, and measurements were conducted on the <2 mm fraction.

| Determination | Unit | Nu'u 1 | Nu'u 2 | Faleālili | Analytical Method |
|---|---|---|---|---|---|
| Sand (>20 µm) | % (*w/w*) | 27.6 | 25.2 | 25.3 | |
| Silt (2–20 µm) | % (*w/w*) | 42.3 | 43.4 | 52.0 | Bouyoucos [34] |
| Clay (<2 µm) | % (*w/w*) | 30.1 | 31.4 | 22.7 | |
| Textural class | - | Clay loam | Clay loam | Silt loam | Australian Soil Texture Triangle |
| $\rho_b$ | g cm$^{-3}$ | 0.886 | - | 0.916 | Blake and Hartge [20] |
| Cumulative infiltration | mm | $F_t = 363.3t^{0.68}$ | - | - | Parr and Bertrand [21] |
| Infiltration rate | mm h$^{-1}$ | $I_t = 204.28t^{-0.35}$ | - | - | Parr and Bertrand [21] |
| Soil pH$_{1:5}$ (soil/water) | - | $5.62 \pm 0.56$ | 6.60 | 4.50 | Rayment and Lyons [35] |
| EC$_{1:5}$ of soil (soil/water) | µS cm$^{-1}$ | $2.92 \pm 0.60$ | - | - | Rayment and Lyons [35] |
| SOC | % (*w/w*) | $3.30 \pm 1.16$ | 12.65 | 3.50 | Walkley and Black [36] |
| Total N | % (*w/w*) | $0.66 \pm 0.21$ | 1.12 | 0.25 | Bremner [37] |
| Soil extractable P | mg kg$^{-1}$ | $2.69 \pm 4.74$ | 28.7 | 14.6 | Olsen et al. [38] |
| Soil exchangeable K | cmol kg$^{-1}$ | $0.46 \pm 0.07$ | 0.77 | 0.45 | MAFF [39] (Method No. 63) |

**Table 2.** van Genuchten [18] model parameters used for representing the relationship between soil water content (expressed volumetrically) and water potential of the Nu'u 1 and Faleālili sites, where $\rho_b$ is (dry) soil bulk density, $\theta_S$ and $\theta_r$ are saturation and residual soil water contents, $\alpha$ and $\eta$ are the fitting parameters of the van Genuchten model (Equations (1)–(3)), R$^2$ is the coefficient of determination, and DUL$_{100}$, LL$_{15}$, and PAWC are laboratory measurements of the drained upper limit (at 100 cm of H$_2$O), crop lower limit (at 15 bar), and plant available water capacity, respectively.

| Site Units | Depth Interval mm | $\rho_b$ g cm$^{-3}$ | $\theta_S$ cm$^3$ cm$^{-3}$ | $\theta_r$ | $\alpha$ - | $\eta$ - | R$^2$ - | DUL$_{100}$ cm$^3$ cm$^{-3}$ | LL$_{15}$ | PAWC mm |
|---|---|---|---|---|---|---|---|---|---|---|
| | 0–150 | 0.876 | 0.6812 | 0.2027 | 0.0522 | 1.5289 | 0.9949 | 0.3945 | 0.2081 | 28.0 |
| Nu'u 1 | 150–300 | 0.885 | 0.6476 | 0.1827 | 0.0574 | 1.4235 | 0.9991 | 0.4019 | 0.2079 | 29.1 |
| | 300–600 | 0.897 | 0.6405 | 0.1913 | 0.0359 | 1.5212 | 0.9973 | 0.4100 | 0.2052 | 61.4 |
| | 0–150 | 0.874 | 0.6824 | 0.2006 | 0.0389 | 1.4374 | 0.9957 | 0.4613 | 0.2269 | 35.2 |
| Faleālili | 150–300 | 0.916 | 0.6397 | 0.1696 | 0.0391 | 1.3899 | 0.9982 | 0.4304 | 0.2054 | 33.8 |
| | 300–600 | 0.958 | 0.6513 | 0.1693 | 0.0313 | 1.4207 | 0.9978 | 0.4528 | 0.2043 | 74.6 |

### 3.2. Crop Measurements and Analyses

Corm yields obtained at Nu'u 1 and Faleālili are shown in Figure 2. At the Nu'u 1 site, overall statistical differences between the control and legume intercropped taro were not significant, and there were no differences between the Erythrina and Mucuna treatments (*p*-values > 0.05). Corm yields obtained at the Faleālili site were only significant between Erythrina intercropped taro and the control ($p < 0.05$), but differences were small.

Differences between the two legume treatments and between Mucuna and the control were not significant ($p > 0.05$). Corm yields obtained at Nu'u 2 in 2020–2021 are shown in Figure 3. Overall, there were significant differences in corm yields between the control and treatments ($p < 0.05$), but amendment type (fertilizer vs. compost) or placement (surface applied vs. incorporated) effects were not significant (*p*-values > 0.05).

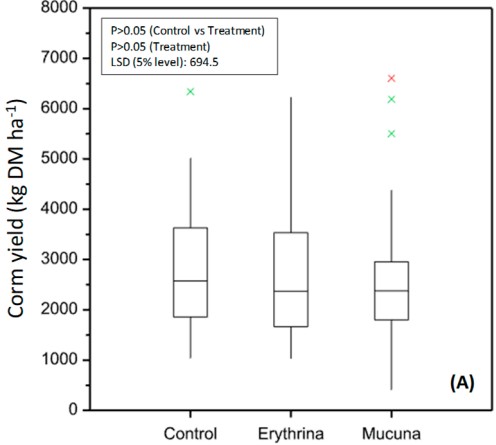 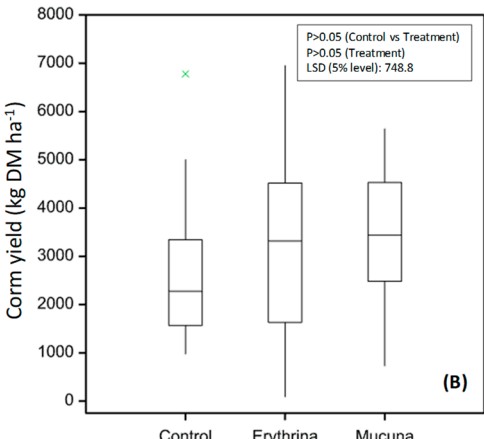

**Figure 2.** Corm yields for the control, *Erythrina subumbrans* and *Mucuna pruriens* treatments were recorded at Nu'u 1 (**A**) and Faleãlili (**B**) in 2018–2019 and are expressed in kg dry matter (DM) per ha. The box spans the interquartile range of the values in the variate ($Q_3$–$Q_1$), with the middle line indicating the median ($Q_2$). Whiskers extend to the most extreme data values within the inner 'fences', which are at a distance of 1.5 times the interquartile range beyond the quartiles (or the maximum value if that is smaller). Individual outliers are identified with a green cross and 'far' outliers (beyond the outer 'fences') are at a distance of three times the interquartile range beyond the quartiles.

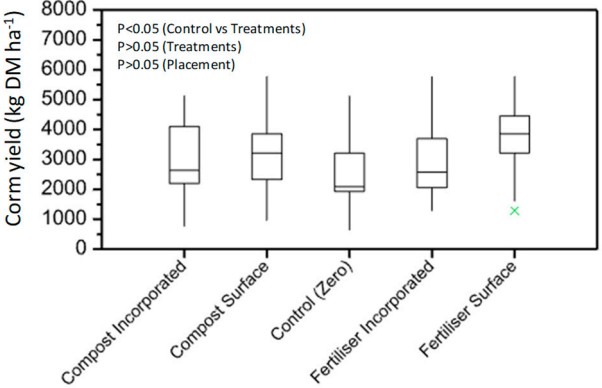

**Figure 3.** Corm yields for the control and composted material based on poultry manure (incorporated and surface application) and compound NPK+S fertilizer (incorporated and surface application) treatments recorded at Nu'u 2 in 2020–2021 expressed in kg dry matter (DM) per ha. Data shown in boxplots are described in Figure 2.

There were no statistical differences in harvest indexes between the control and treatments or between treatments at any of the sites (*p*-values > 0.05). Averaged across treatments, harvest indexes were 53.1% (*w/w*) at Nu'u 1 and 58.7% (*w/w*) at Faleãlili. At Nu'u 2, mean harvest indexes were between 57.8% and 60.6%, and differences between amendment type or placement (surface vs. incorporated) were not significant (*p*-values > 0.05). Harvest indexes were within the range reported in earlier studies (e.g., [40,41]). Agronomic efficiency calculations for this site were 11.3 kg DM kg$^{-1}$ N, 38.9 kg DM kg$^{-1}$ P, 10.9 kg DM kg$^{-1}$ K for fertilizer treated crops, and 12.3 kg DM kg$^{-1}$ N, 62.2 kg DM kg$^{-1}$ P, and 20.4 kg DM kg$^{-1}$ K for compost treated crops, respectively.

Table 3 shows the elemental nutrient composition used to derive nutrient off-take which was subsequently used with DM yield data to provide field scale nutrient balance estimates. Figures 4 and 5 show the estimated nutrient off-take for the three experimental sites based on the taro corm yields reported in Figures 2 and 3, and the average elemental composition of taro corms presented in Table 3. Given that treatment differences in yield encountered at Nu'u 1 and Faleãlili were not significant (except for the small yield difference between the controls and Erythrina at the Faleãlili site), nutrient off-take data were consolidated into single figures per site. For Nu'u 2, nutrient off-take data are presented by amendment type, as other treatment effects (amendment placement) on yield were not significant (*p*-values > 0.05).

**Table 3.** Elemental composition of taro corms used to estimate nutrient off-take at harvest. Values were compiled and averaged from earlier experiments conducted at the sites that used the same taro varieties [13,26]. SD is standard deviation. Mean taro corm DM was 35.67 ± 4.51% (*w/w*).

| Element | Unit | Mean Concentration ± SD |
|---|---|---|
| Nitrogen, N | %, *w/w* (dry basis) | 0.76 ± 0.142 |
| Phosphorus, P | %, *w/w* (dry basis) | 0.24 ± 0.012 |
| Potassium, K | %, *w/w* (dry basis) | 1.45 ± 0.289 |
| Calcium, Ca | %, *w/w* (dry basis) | 0.10 ± 0.025 |
| Magnesium, Mg | %, *w/w* (dry basis) | 0.15 ± 0.021 |

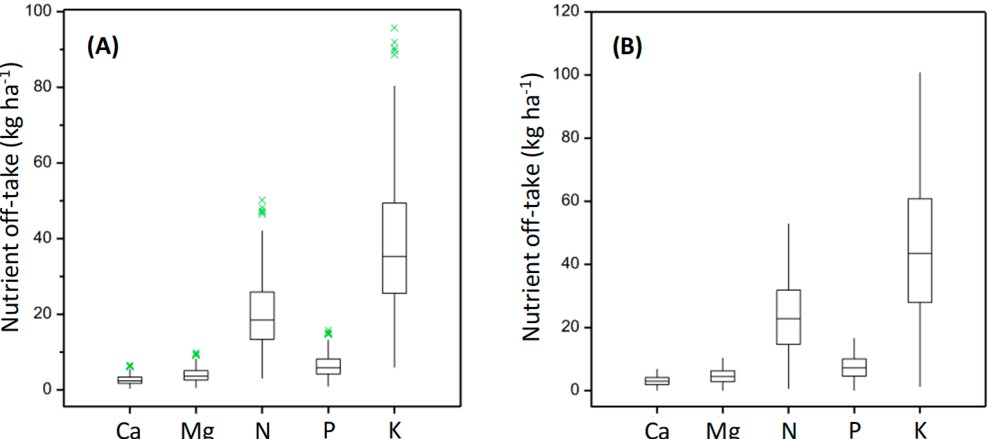

**Figure 4.** Estimated nutrient off-take in taro corms at Nu'u 1 (**A**) and Faleãlili (**B**) in 2018–2019. Data shown in boxplots are as described in the caption of Figure 2.

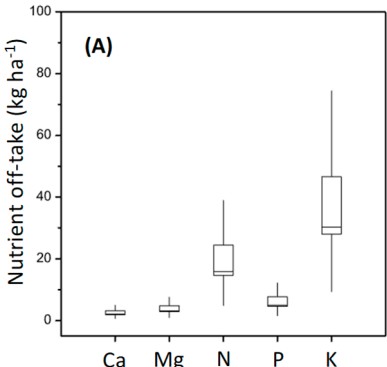 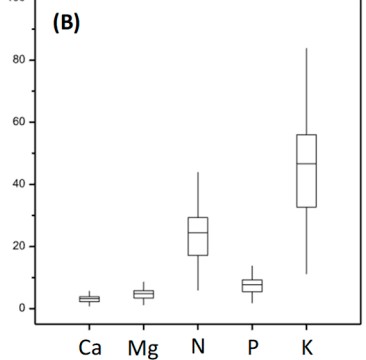 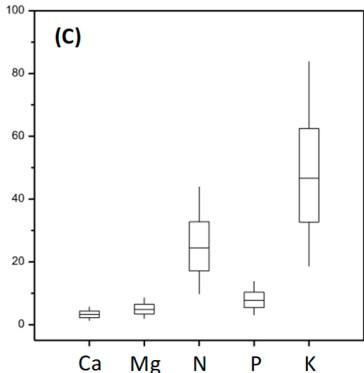

**Figure 5.** Estimated nutrient off-take in taro corms at Nu'u 2 in 2020–2021. (**A**) control (zero amendment), (**B**) composted material based on poultry manure, and (**C**) compound NPK+S fertilizer applied at 50 g (product) per plant. Data shown in boxplots are as described in the caption of Figure 2.

The data presented in Figures 4 and 5 were subsequently used to provide field scale nutrient balance estimates for five major elements (Figure 6A–C). Based on the assumptions made in the analyses, these estimates resulted in negative balances across all five nutrients when legumes were intercropped with taro, including N. The apparent N, P, and K surplus estimated at Nu'u 2 for fertilizer- and compost-treated taro did not correspond with the yields recorded at this site. This observation suggested that corm yields were more constrained by factors other than nutrition (importantly weed control). The apparent nutrient surplus simply reflected the poor use efficiency of applied nutrients, whether as compost or fertilizer. Average nutrient off-take ratios were, approximately, 6:2:11:1:1 (N:P:K:Ca:Mg), which may be used as guidance for fertilizing taro crops if a nutrient replacement strategy was to be adopted.

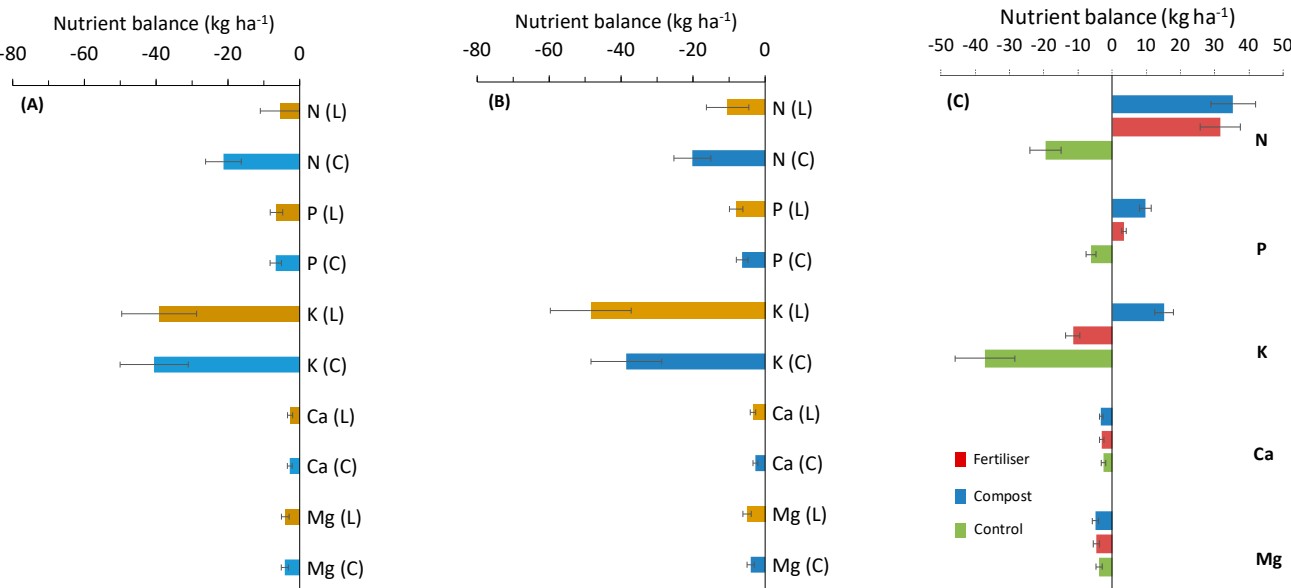

**Figure 6.** Estimated field scale (macro) nutrient balance at three experimental sites as affected by nutrient management practice. From left to right, (**A**) Nu'u 1 (2018–2019), (**B**) Faleālili (2018–2019), and (**C**) Nu'u 2 (2020–2021). In (**A**,**B**), the letters (L) and (C) following the nutrient symbol denote 'legume' intercropping (light brown columns) and the 'control' (blue columns), respectively. In (**C**), 'fertilizer' is compound NPK+S fertilizer and 'compost' is a composted material based on poultry manure.

## 4. Discussion

### 4.1. Soil Effects

Results derived from measurements of soil hydraulic properties and local rainfall and evapotranspiration records suggested that it may be possible to grow taro slightly off the standard growing window for the crop, which is between late August and March. A shift in the timing of planting toward the rainy season would allow a second taro crop to be established between December and February. This can be achieved by relay intercropping a first taro crop, established within the standard planting window (late August-early September) with a second taro crop established at the beginning of the rainy season. This second taro crop would then be harvested between August and November, depending upon its planting time. Therefore, the first and second taro crops would co-exist during part of the rainy season and potentially between August and November, again depending upon their relative planting and harvest times. The establishment of the second taro crop early in the rainy season would be advantageous to reduce the risk of water stress in the first crop as it approaches maturity. Equally, competition for water at later phases of development of the second taro crop (when the first crop is re-established around August-September) may also be minimized. Successful implementation of the proposed double-taro cropping strategy would allow harvesting two crops in 12 months instead

of 19 months under current practice. However, such an approach must be supported by appropriate fertilization and crop protection programs, and adequate soil husbandry that maximizes soil water conservation [42,43].

Virtual trials in the APSIM (Agricultural Production Systems Simulator, https://www.apsim.info/ (accessed on 30 July 2022)) using the taro module [44,45], coupled with water balance modeling approaches; for example, using SWIM3 [46], could be employed to assist farming system optimization. Field-based experimentation is required to validate this recommendation. This is a prerequisite for successful adoption and implementation in commercial scale farming.

*4.2. Crop Effects*

The average corm yield at Nu'u 1 (2730 kg DM ha$^{-1}$) was 1370 kg DM ha$^{-1}$ lower than the national average ($\approx$4100 kg DM ha$^{-1}$) recorded over the five-year period prior to these experiments [47]. The attainable corm yield in Samoa has been estimated at 6150 kg DM ha$^{-1}$ (FAO, https://www.fao.org/3/ad513e/ad513e0c.htm#bm12.1 (accessed on 12 August 2022)). For rainfed cropping systems, the attainable yield may be regarded as an approximation of the water-limited yield [4,48]. From these results and based on previously published data, it can be inferred that the yield gap between actual (field measured) and attainable yields was approximately 3400 kg DM ha$^{-1}$. At the Faleālili site, the yield gap between the average legume intercropped taro (3340 kg DM ha$^{-1}$) and the national average was estimated at 750 kg DM ha$^{-1}$, about 2800 kg DM ha$^{-1}$ compared with the attainable yield.

Results also highlighted the limited efficacy of legume intercropping as a strategy to increase the productivity of taro. Therefore, the use of legumes may be discouraged in soils with declined fertility, such as the Nu'u 1 site, which exhibited rather low soil extractable P levels (Table 1). When intercropping legumes with taro, the following management aspects may need to be considered:

- Intercropping may lead to an increased risk of water stress; particularly, if due to the selected planting date, the crop cycle extends into the 'dry' season (e.g., [41,49,50]).
- Legumes will likely reduce the availability of soil P to the growing taro crop in soils that are under-supplied with P; for example, Olsen's P below 20 mg kg$^{-1}$ [51], as legumes are known to take up substantial amounts of P [52]. Similar effects may be encountered with other nutrients (e.g., K, Ca, Mg) if their levels in soil are below critical values for growing taro.
- Low soil P supply will likely reduce N uptake by the taro crop because of significant N × P interaction on N uptake (regardless of N being available to the taro crop as a result of N fixation by the legume) [53].
- Soil application of P, whether as a fertilizer or organic amendment, should account for P uptake by the legume (which may result in temporary immobilization of P in legume biomass) when this is intercropped with taro. The same consideration applies to other key nutrients (e.g., K, Ca, Mg) used by legumes in fairly large quantities [54].
- Target fertilization strategies to meet nutrient requirements of both legume and taro crops so that N fixation and soil N supply to the growing taro are not compromised, and plant uptake of other nutrients is not impaired. Careful optimization of the system will be required to ensure that increased water use by legumes (due to the likely increased biomass in response to applied fertilizer) does not hinder water (and, therefore, nutrient) uptake by the growing taro (co-limitation) [55–57].

On average, yields obtained in the amendment-treated crop at Nu'u 2 were approximately 950 kg DM ha$^{-1}$ lower than the national average and about 40% to 50% lower than the attainable yield for Samoa. The attainable yield can only be achieved through the skillful use of the best available technology [4], which does not appear to be represented by the use of legumes as a strategy for replacing fertilizer N. Results obtained at Nu'u 2 also suggested that the overall agronomic performance of amendment-treated crops was more constrained by management factors other than plant nutrition. Technical officers

responsible for the field experiments were unable to visit the site while in the COVID-19 lockdown. This meant that the experiments were unattended for extended periods and that routine crop protection measures could not be appropriately performed. Lack of timely weed control had adverse effects on yield and nutrient recovery in crop biomass. Yields at Nu'u 2 were within the range of yields recorded at Nu'u 1, despite that this site had no nutrients applied (other than N derived from intercropped legumes). The challenges faced by technical officers and field personnel during the 2020–2021 season were mostly outside their control. However, this made them aware of the need for 'good' overall crop husbandry if high-performing crops were to be produced. There appeared to be scope to increase actual (field measured) yields by about 3000 kg DM ha$^{-1}$ and the national average yield by about 2000 kg DM ha$^{-1}$ should best management practices for nutrients and crop protection were to be implemented. Farming systems modeling approaches that rely on the use of the APSIM [44,45] may be applied to simulate the combined effects of balanced nutrition and soil water availability on crop performance. The APSIM can be also used to identify management scenarios that will likely narrow yield gaps, and increase nutrient and rainfall use efficiency in the context of climate change [45,58–60].

Nutrient balance estimates were negative across all five elements when legumes were intercropped with taro, including for N. The apparent N, P, and K surplus estimated at Nu'u 2 for fertilizer- and compost-treated taro did not correspond with the yields recorded at this site. This suggested that corm yields were more constrained by factors other than nutrition (importantly weed control). The apparent nutrient surplus at this site simply reflected poor use efficiency of applied nutrients, whether as compost or fertilizer. Average nutrient off-take ratios were, approximately, 6:2:11:1:1 (N:P:K:Ca:Mg), which may be used as guidance for fertilizing taro crops if a nutrient replacement strategy was to be adopted.

## 5. Synthesis: Nutrient Management Framework

A nutrient management framework is proposed here to guide nutrient recommendations for taro production systems (Figure 7). Recommendations for the application of N can be derived from the yield-to-N response relationship and the price ratio; that is, the unit price of N relative to the unit price of the crop [61]. Recommendations for the application of P, K, Ca, and Mg are based on a nutrient replacement strategy. Nitrogen inputs to a crop can be optimized by deriving the most economic rate of N from the yield-to-N response relationship (Equation (7)) (after [24,62,63]):

$$Y = a + bx + cx^2 \tag{7}$$

where $a$, $b$, and $c$ are regression coefficients, $x$ is N application rate, and $Y$ is yield. The lowest N application rate at which the maximum $Y$ is obtained can be obtained by equating the first order differential to zero (Equations (8) and (9)):

$$\frac{dy}{dx} = b - 2cx' = 0 \tag{8}$$

∴

$$x' = \frac{b}{2c} \tag{9}$$

where $x'$ is the lowest N application rate at which the maximum Y (Y$_{\text{MAX}}$) is obtained, and where $x < x'$. The N rate that corresponds with $x'$ is referred to as N$_{\text{MAX}}$. The most economic rate of N (MERN) is obtained when the differential is equated to the price ratio ($P_R$), as follows:

$$b = 2cx' = P_R \tag{10}$$

and

$$P_R = \frac{P_N}{P_C} \tag{11}$$

$\therefore$

$$MERN = \frac{b - P_R}{2c} \tag{12}$$

where $P_R$ is the price ratio, $P_N$ is the price of N ($\$$ kg$^{-1}$), $P_C$ is the price of the harvested crop product (taro corm, $\$$ kg$^{-1}$), and MERN is the most economic rate of N (kg ha$^{-1}$) for a given price ratio $P_R$.

The price ratio ($P_R$) is equivalent to the breakeven ratio and indicates the extra return of the crop produced that just covers the extra unit of N added. At this point, the economic return from N applied as fertilizer or amendment is maximized (the cost of any additional N is greater than the value of the extra crop yield produced). When $P_R$ increases, for example, as a result of increased N price, MERN is concurrently reduced to allow for the same rate of economic return from applied N. Nitrogen rates below MERN will result in economic penalties as crop yield will be limited by N supply. Nitrogen rates above MERN will also reduce the economic return from applied N, as any yield increment achieved above this rate will become proportionally smaller. The yield achieved with an N input equivalent to MERN is referred to as $Y_{MERN}$. At N rates up to and including MERN, there is a roughly constant amount of residual soil N that may be lost by the processes of leaching or denitrification. Above this rate, the risk of environmental losses of N losses increases in a non-linear fashion due to a proportionally larger surplus of applied N [64,65].

Since responses to applied N are site (edaphic-related effects) and season (climate-related effects) specific, a family of response curves constructed over multiple years at a given location will offer the required confidence to make N decisions at such location. The optimum economic rate can be adjusted using the yield-to-N response (developed with historical data) and year specific $P_R$. This approach forms the basis for making N recommendations under the proposed nutrient management framework. Further work needs to be undertaken to establish yield-to-N response relationships at key locations in Samoa, in order that trusted N management advice can be rolled out to local farmers to inform management decisions. The experimental station of the Samoan Ministry of Agriculture and Fisheries at Nu'u (https://maf.gov.ws/ (accessed on 26 October 2022)), together with the Samoa Farmers Association (https://pacificfarmers.com/ (accessed on 26 October 2022)), and the technical assistance from SROS, may offer opportunities for such technical and extension work to be conducted.

Recommendations for the application of P, K, Ca, and Mg should be based on a replacement strategy. Soil nutrient indexes applicable to Samoan soils need to be developed. Further, critical (or target) indexes below or above which agronomic and profitable responses are or are not likely to be encountered also need to be determined. For soils in which a given nutrient index is above the target index, the application of that nutrient may be omitted as there would be no agronomic or economic incentive to do that so. Specifically for P, its application to high P index soils needs to be avoided to reduce the risk of environmental losses (from soil to water) [66,67]. By contrast, for soils with a nutrient index below its target index, the application of that nutrient cannot be avoided if yield and economic return from applied nutrients are to be optimized. For these low-index soils, there should be a long-term policy aimed at building up nutrient deficiencies allowing for progressive correction of indexes toward target levels. For soils with nutrient levels already at the target index, the objective should be to maintain it and only apply nutrients on a replacement basis. The application of a given nutrient to soils that are at the target index may be omitted in some years when, for example, $P_R$ is too narrow (the nutrient price is too high or crop price is too low, or both). Soil testing for determination of P, K, Ca, and Mg indexes may be conducted at regular time intervals (e.g., 3–4 years), along with an annual fertilization plan that takes an account of the crop rotation developed afterward. For N, soil testing may be omitted, as obtaining reliable analytical results while being able to accurately quantify soil N supply rates under the local Samoan conditions may be cumbersome. Soil sample preservation while in transit from the field to the laboratory and the storage time and conditions before they are analyzed will significantly influence available N fractions [68]. Therefore,

the reliability of data derived from soil analyses may be compromised. Derivation of the optimum N application rate (MERN) from the yield-to-N response is considered to be a more reliable approach to formulating N recommendations, and it will also help reduce costs associated with soil analyses.

The framework highlights the focus of the nutrient management strategy, namely (*i*) nutrient build-up (productivity), (*ii*) maintenance (productivity, environmental), and (*iii*) omission of application in some years (environmental) if nutrients were already at an agronomically satisfactory level. Thus, the framework accounts for the primary factors upon which the recommendation for specific nutrients should be based (response, replacement). The approach establishes a criterion for determining the 'right rate' and emphasizes that all 4*R* (*R*ight source, *R*ight rate, *R*ight time, *R*ight place) Nutrient Stewardship Principles [69,70] must always be observed.

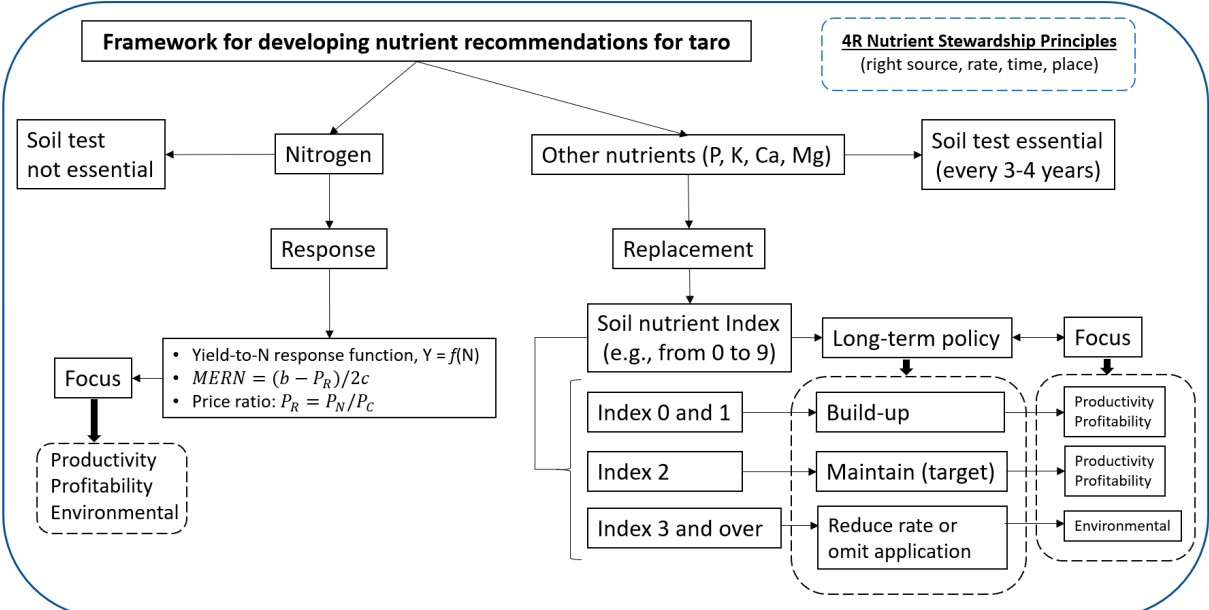

**Figure 7.** A conceptual framework to support the development of nutrient recommendations for taro production systems. The yield-to-nitrogen (N) response function assumes a quadratic plateau relationship [24,71]; MERN is the most economic rate of N (Equation (12)) and can be derived directly from the response curve; and $P_R$ (Equation (11)), $P_N$, and $P_C$ are the price ratio, price of N, and price of the crop, respectively [24,62]. The numerical scale used to define soil nutrient index (from 0 to 9) is given as an example and it may be modified to suit specific requirements. The 4*R* Nutrient Stewardship Principles [69,70] must always be observed.

Table 4 shows how the soil nutrient index concept (Figure 7) may be translated into a fertilizer recommendation. For this, P is used as an example, and it is assumed that the target corm yield is equivalent to the attainable yield ($Y_a$ = 6150 kg DM ha$^{-1}$). A similar procedure may be then applied to the case of K, Ca, and Mg. However, the information presented in Table 4 requires validation and it, therefore, may not be used to formulate fertilizer recommendations. This information is presented for the sake of providing a worked example for key nutrients. Further work needs to be undertaken to validate these concepts.

In Table 4, if soil P index was 3, the recommended rate could be omitted (or reduced) in some years when, for example, the price ratio was narrow, or soil extractable P levels were above the middle point of Olsen's range (>35 mg kg$^{-1}$).

**Table 4.** Formulating fertilizer recommendations based on the soil nutrient index concept presented in Figure 7. Phosphorus (P) is used as a worked example, and it is assumed that the target corm yield is equivalent to the attainable yield ($Y_a$ = 6150 kg DM ha$^{-1}$). Olsen's P ranges and soil P indexes were adapted from DEFRA [72].

| Olsen's P | Soil P Index | Strategy | Recommended Application Rate | |
|---|---|---|---|---|
| (mg kg$^{-1}$) | - | - | (kg P ha$^{-1}$) | (kg P$_2$O$_5$ ha$^{-1}$) |
| 0–9 | 0 | Build up | 28 | 65 |
| 10–15 | 1 | Build up | 22 | 50 |
| 16–25 | 2 | Maintain | 15 | 35 |
| 26–45 | 3 | Reduce/Omit | 9 | 20 |
| 46–70 | 4 | Omit | 0 | 0 |
| 71–100 | 5 | Omit | 0 | 0 |

## 6. Conclusions

This work investigated field scale nutrient cycling in rainfed taro production systems and demonstrated the importance of nutrient budgeting for long-term soil fertility management. This work made it possible to quantify field scale nutrient balances. A framework for developing nutrient recommendations for taro production systems, informed by data collected during these experiments, was presented and discussed. The main conclusions from this work are summarized below:

(1) *Soil fertility and soil organic carbon (SOC)*: there were no significant changes in any of the measured soil parameters, as determined before and after growing taro. Significantly lower SOC and nutrient levels and soil pH at Nu'u 1 and Faleālili compared with Nu'u 2 were attributed to the relative number of years under continuous cropping at these sites (Nu'u 2 was a newer site with less than 3 years of cropping). Differences in SOC, soil pH, and nutrient levels (especially total N and Olsen's P) between sites suggested high vulnerability to soil fertility decline; particularly when soils are used for cropping without significant C and nutrient inputs.

(2) *Legume intercropping*: the assumed amount of N supplied via fixation (range: 40 to 60 kg ha$^{-1}$) appeared to be insufficient to meet the taro crop demand for N. If temporary immobilization of P in legume biomass due to P uptake is significant, soil P availability to the growing taro crop may be concurrently reduced. This can compromise N and K uptake, and affect crop water use (rainfall use efficiency). These effects may be exacerbated in lower-fertility soils and soils with low water-holding capacity. If legumes were to be intercropped with taro, the fertilization program should account for the nutrient demands of both crops. However, this will require careful optimization of the system to ensure that increased water use by legumes (due to increased biomass in response to applied fertilizer) does not limit water and nutrient uptake by the growing taro crop.

(3) *Soil mineral and organic amendments*: overall, the nutrient balance was negative when legumes were used. Apparent surpluses of N, P, and K at Nu'u 2, when either compost or fertilizer was used, were explained by low corm yields and, therefore, poor nutrient use efficiencies (recovery in corm biomass) and lack of weed control over the season. Despite this, corm yields were higher in amended (compost or fertilizer) treated taro compared with legume intercropping. The country's attainable yield (6150 kg DM ha$^{-1}$) may not be achieved without the balanced application of nutrients and proper weed control.

(4) *Agronomic performance of taro*: corm yields were lower than the national average and the estimated yield gap (difference between actual and attainable yields) was wide. Surface application of nutrients reported similar yields to soil incorporation, which suggested that inadequate weed control and different nutrient loss mechanisms may be driving such effects. If best nutrient and crop protection management practices

could be implemented, actual corm yields and the national average corm yield could be increased by approximately 3000 and 2000 kg DM ha$^{-1}$, respectively.

(5)  *Nutrient guidelines*: Under the proposed nutrient management framework, N recommendations should be derived from the yield-to-N response function. For other nutrients (P, K, Ca, Mg), recommendations should be based on replacement. Knowledge of the yield-to-N response relationship will enable derivation of the most economic rate of N (MERN) to ensure the economic return from applied N is maximized. The replacement strategy will require the development of soil indexes. These indexes can be used to define the long-term (field scale) nutrient management policy and help practitioners make better nutrient decisions. This long-term nutrient management policy needs to be informed by soil analyses to determine whether soil nutrient levels need to be built up or maintained. Application of a given nutrient could be omitted in some years if the soil index denotes a satisfactory level. It is envisaged that the adoption of this framework will increase soil nutrient security and resilience of taro production systems.

**Author Contributions:** Conceptualization, B.C.T.M. and M.J.W.; methodology, D.L.A. and A.U.; formal analysis, D.L.A. and B.C.T.M.; investigation, A.U. and D.L.A.; resources, J.P. and J.K.; data curation, D.L.A. and A.U.; writing—original draft preparation, D.L.A., B.C.T.M. and U.S.; review and editing, U.S. and J.R.F.B.; supervision, D.L.A.; project administration, B.C.T.M. and S.T.; funding acquisition, B.C.T.M. All authors have read and agreed to the published version of the manuscript.

**Funding:** This research received financial support from the Australian Centre for International Agricultural Research (ACIAR, Australian Government, https://www.aciar.gov.au/ (accessed on 26 October 2022), Project ID SMCN/2016/111), and the Australian Science and Technology for Climate Partnership (Department of Foreign Affairs and Trade, Australian Government, https://www.dfat.gov.au/ (accessed on 26 October 2022). Further details about the work jointly conducted by CSIRO, its partner organizations, and ACIAR in Pacific countries and territories can be found at https://research.csiro.au/pacsoils/ (accessed on 26 October 2022), including the development of an open access soils information system known as The Soils Portal: https://research.csiro.au/pacsoils/our-research/digital-soil-portal/ (accessed on 26 October 2022).

**Institutional Review Board Statement:** Not applicable.

**Informed Consent Statement:** Not applicable.

**Data Availability Statement:** Data associated with this study can be requested from Dr. Diogenes L. Antille (dio.antille@csiro.au). Background information is available at https://research.csiro.au/pacsoils/ (accessed on 26 October 2022) and https://research.csiro.au/pacsoils/our-research/digital-soil-portal/ (accessed on 26 October 2022).

**Acknowledgments:** The authors acknowledge the operational support provided by The Scientific Research Organisation of Samoa (SROS, https://sros.org.ws/ (accessed on 26 October 2022)), The Ministry of Agriculture and Fisheries of Samoa (MAF, https://maf.gov.ws/ (accessed on 26 October 2022)), Manaaki Whenua Landcare Research (New Zealand), The Pacific Community (SPC, https://www.spc.int/ (accessed on 26 October 2022)), extension officers and technical personnel from MAF and SROS, and local taro growers. Help received from G. McLachlan and S. Tuomi (CSIRO Agriculture and Food, Canberra), and Angelika Tugaga (SROS) is gratefully acknowledged. Comments and suggestions by anonymous reviewers from CSIRO Agriculture and Food (Canberra, Australia), and the Editor and Reviewers of this journal are appreciated.

**Conflicts of Interest:** The authors declare no conflict of interest. The funders had no role in the design of the study; in the collection, analyses, or interpretation of data; in the writing of the manuscript, or in the decision to publish the results.

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
