# Peer review of "Toward Soil Nutrient Security for Improved Agronomic Performance and Increased Resilience of Taro Production Systems in Samoa"

_soilsystems, doi:10.3390/soilsystems7010021_

Round 1

Reviewer 1 Report

The article presents a study of fertilization for Taro in Samoa. The investigation is relevant and the proposition is adequate. The article is well written and discussed, and presents plausible conclusions, and does not present plagiarism. However, the authors should better present the proposed objective for the study and have many keywords.

I do not consider the topic as original because organic fertilization is widely used in the world but the approach taken by the authors is relevant to the area because it discusses the change from the traditional model of fertilization to organic fertilization.    This publication adds the use of cover crops as a source of nutrients for Taro in two different locations and brings the balance of nutrients in the systems adopted.

The authors did not use the same treatments at the different sites, which makes comparison between them difficult, and they could have used more time to verify that the results are resilient as mentioned in the title of the article.
The authors did not present the concentration of nutrients in the leguminous plants and poultry manure, making it difficult to verify the balance of nutrients in the adopted cropping systems.
The authors do not need to address the issue of water infiltration into the soil. Soil water infiltration is not the objective of the study.

The conclusions are too extensive, and run away from the objective of the study which is the nutritional part of the Taro plant and its fertilization as a function of the use of different sources of organic and non-organic fertilizers.
The conclusions do not refer to the title of the article. For, the food security approach was not studied since that is too broad for the purpose of this article.

Author Response

Dear Reviewer,

Thanks very much for taking the time to review our manuscript and for your valuable comments. We hope our corrections are acceptable, and if you have any concerns, please let us know so we can address them accordingly.

Please find attached an itemised set of responses to each of the comments.

With thanks and kind regards,

Dr Diogenes Antille

CSIRO Agriculture and Food

Canberra, Australia

Reviewer 2 Report

The manuscript is interesting. It poses a serious local problem in Saoma and with a real focus on the use of the soil and the deficiency of nutrients caused by agronomic use (in particular taro production). The work is well structured.

 In general:

 I recommend revising and reducing the text of each figure. The details of the figures should be located in the body of the manuscript.

 Several sections must be separated into paragraphs, very long texts and sometimes continuous reading is difficult.

In particular:

 In section 2.2 Experimental sites: Treatments and plots are described in a confusing way; you must rewrite these details. You should separate the section into paragraphs to improve interpretation.

 Line 134: “taro outside the normal window for the crop”, change by: taro outside the normal season for the crop.

 Figure 7 is very important. It should be considered in the objectives of the work and reinforce its importance in the introduction of the manuscript

Author Response

(The authors gave the same response as above.)

Reviewer 3 Report

This paper to examines the effect of intercropping legumes and nutrient amendments on taro yield. There are deficiencies in the methodology. There appear to be priors that were not accounted for in the experimental design or rationally explained why they were not included. There appear to be methodological constraints, though outside of the authors’ control, that impacted the strength and results of the experiment, which can not be ignored. My sense is there was an experiment, and because of these constraints (i.e., Covid impacts), the results were not robust as intended, but a paper was still developed. The outcome is the paper is not always cohesive with the experimental results not providing insight into the dynamics of the system.

Introduction: the general impact of agriculture on nutrient flows and the impact on yield gap is presented. Is there literature on the impact specifically on taro production and economic losses? It is not clear the breadth of the problem requiring improved soil nutrient management. I’m sure it is warranted, can the authors provide the quantitative supporting information.

Line 111: Did the authors conduct a nutrient analysis on the manure/compost amendments?

Line 116: Was the surface amendment added at the time of planting? If so, what was the risk of runoff of the nutrient application during rain events? Was this mitigated with any actions?

Line 134-135: What are the implications of “growing outside the window,” and how does the second simultaneous cropping relate to issues or problems presented in the Introduction?

Line 141-174: can the authors provide the reader in the Introduction the rationale via hypotheses or expectations of the impact of soil water content and water potential on the expected outcomes?

Line 194: How did the authors determine the nutrient input for compost treatments?

Line 199: Did you conduct a sensitivity analysis to evaluate the impact of choosing 50 kg N ha/yr as a contribution rate?

Line 247: Why “but” this is the same response as the legume plantings.

Line 313-225: This finding is not aligned with the original purpose of the experiments presented in the Introduction. Further, this is a hypothesis that should be tested. A hypothesis should not be presented as the first discussion point. Finally, the recommendations of crop timing and rationale is not based on experimental results and are theoretical here.

Line 337-347: Why are the yields lower? The authors do not fully address this question.

Line 353 – 374: Why weren’t these considered during experiment development? If these are known priors why didn’t the experiment take these into consideration, or why were they not addressed in the Introduction explaining why they were not taken into consideration.

Line 401: It appears that weed control may have negatively impacted the ability to detect differences in yield output based on nutrient loadings. From that perspective, how is the experiment didn’t fail since there is a variable unaccounted for (I respect this was outside of the authors’ control because of covid, regardless, it is obviously impacted the experiment)

Line 406 -517: In the traditional IMRDC paper format, the presentation of methods and results (Tables and Figures) are not typically presented in the Discussion section. Also, how does this framework relate to any of the experimental analytical results, it appears the lack of detected treatment effects resulted in this additional framework approach.

Line 477-485: Soil testing may be omitted? If soil testing for nutrients is a moot point because of methodological constraints, why implement this as part of the experiment, to say  “Soil testing may be omitted”. Again, if this was a prior, why wasn’t it taken into consideration in the experimental design and implementation?

Line 520: It is not clear at all the results support this conclusion

Line 538: This outcome should have been known without the experiment. See Authors’ list at line 353-374

Author Response

(The authors gave the same response as above.)
